# Associations between Work Resources and Work Ability among Forestry Professionals

Hannu Pursio [1], Anna Siukola [2], Minna Savinainen [3], Hanna Kosonen [1], Heini Huhtala [1] and Clas-Håkan Nygård [1,*]

1   Faculty of Social Sciences, Tampere University, 33014 Tampere, Finland; hannu.pursio@tuni.fi (H.P.); hanna.kosonen@tuni.fi (H.K.); heini.huhtala@tuni.fi (H.H.)
2   Faculty of Medicine and Health Technology, Tampere University, 33014 Tampere, Finland; anna.siukola@tuni.fi
3   Finnish Institute of Occupational Health, Työterveyslaitos, 33100 Tampere, Finland; minna.savinainen@ttl.fi
*   Correspondence: clas-hakan.nygard@tuni.fi

**Abstract:** Globalization and structural changes in forestry have changed the content and operating practices of timber harvesting. Furthermore, digitization and new forms of work organization have changed work characteristics, requirements and resources. The importance of knowledge and competence, and the management of new technology, are emphasized more. The purpose of this study was to find out how work resource factors are related to the work ability of forest machine entrepreneurs and drivers. The research material was collected in 2018 through an online survey involving 322 professionals in the timber harvesting industry, 87 forest machine entrepreneurs and 235 forest machine drivers. The Mann-Whitney U-test and logistic regression analysis have been used for statistical analysis. Effective work organization and social support from co-workers, as well as the perceived meaningfulness of one's work, were resources that increased the likelihood of good work ability. Based on our results, good management may enhance work resources, and by developing work it is possible to support employee ability amid the pressures of change inherent to a competitive commercial environment and new forms of work.

**Keywords:** work resources; work ability; forest industry; timber harvesting; restructuring

## 1. Introduction

Global structural change in the forest industry has altered the operating environment for timber harvesting, where three key stages of change can be seen. Firstly, the starting point is the global restructuring of the forest industry [1,2]. Secondly, as a result of restructuring, operational responsibility for timber harvesting has shifted from forest industry companies to specialized forest service companies [3,4]. Thirdly, at the same time, significant technological developments, of which digitalization is central, have taken place in the industry [5,6]. As a result of these key changes, work has been reorganized to correspond to these global developments [7,8]. The new forms of work and new business models and technological development of timber harvesting have changed work organization and management practices, as well as the work itself [9–11]. New responsibilities and practices mean independent planning and the ability to make independent decisions in the work of forest machine entrepreneurs (entrepreneurs) and machine operators (drivers), as well as receiving and producing digital information for forest industry information systems [12]. As a result of technological development, forest machines are increasingly becoming independently operating units in an intelligent data network [5]. From a business perspective, digitization means changes in operating practices brought about by digital technology that aim to increase performance, operational efficiency, accessibility and cost savings [5,13]. Developments in timber harvesting are moving towards larger harvesting

companies and wood supply chains, as well as towards greater efficiency and more versatile services [4,14,15]. As a consequence, the successful operation and performance of companies is more than before tied to individuals, employee motivation and work ability, and ability to adapt to changing of work [16]. With work becoming more subjective and autonomous, employee responsibility for the outcome of the work will increase [17].

The increase in work-related competence and decision-making ability, as well as time pressures, are associated with increased work efficiency [18,19]. Changes in work and work organization practices have made it necessary also for forestry to adapt to a new environment in terms of work content [4,5,20]. Technological developments and the increase in work efficiency, as well as data management competence, have increased mental and cognitive workload [18,21]. Electronic mobile information technology has streamlined and accelerated work processes and changed work organization practices [22,23]. In previous studies, structural changes in work have been found to be associated with increased workload and reduced employee well-being, as well as perceived impaired work ability [24,25].

Work ability is a social concept and construction [26,27]. It can be viewed from the perspective of both assessment and its promotion as a multidisciplinary concept [28,29]. Work ability is also seen as a dynamic process that changes as an employee's resources change. The concept of work ability also acquires new content as work and the operating environment and society change. Work ability is determined by how the psychosocial, physical and organizational work environment enables employees to use their resources to achieve work goals [29,30]. Good personal resources, such as diverse professional skills and motivation, as well as adequate health, increase resilience and support work ability as the operating environment and the nature of work change [29,31]. It is essential to identify the changes and impacts of work when the goal is to anticipate the resources needed and to promote employee work ability through workplace actions [29,32,33].

The theoretical framework of the present study is the Work Ability House model [34], which is based on a multidimensional concept of work ability [28,34,35]. The model consists of four different dimensions (levels), which are health, competence, values and attitudes, and work characteristics. The most important factors related to work ability are health and work characteristics [36]. According to the model, work ability factors included in work characteristics are physical working conditions and the psychosocial work community, as well as management and organizational operations. In addition to these characteristics of work, the content and requirements of work form a whole that is affected by technological development and globalization, as well as the surrounding society [35,36]. Work ability is a balance between work and work resources, where balance means that the individual's work ability is sufficient to achieve work goals. All levels of the model interact with one another [35,36]. In the present study, the work ability of timber harvesting professionals is examined from the perspective of resources related to work and working conditions in their own operating environment [29,37,38].

Work characteristics can be divided into two categories, regardless of occupation or organization: work demands and work resources [39]. Work resources refer to the physical, mental, social or organizational characteristics of work that balance and buffer the strain produced by work demands. Such resources include, e.g., work organization, opportunities to influence one's own work, support from colleagues at work, and the meaningfulness of work [40,41]. In this study, the meaningfulness of work refers to employees 'experience that timber harvesting work has a broader social significance in the value chain of sustainable development in the forest industry' [40].

Resources reduce the negative effects of work demands and support employees in personal professional development and in achieving work goals [39,42]. Previous research has found that resources related to leadership, the work community, and job content, such as good work organization [43], social support in the workplace [44], and the meaningfulness of work [45,46] support employee performance and work ability and increase organizational productivity [47].

The aim of this study was to look at the association between work resources and work ability and to produce new information, on the basis of which workplaces can identify and enhance those work resources that support work ability while taking into account changes to work, its new forms, and the corporate environment.

## 2. Materials and Methods

### 2.1. Material

The data is from a 2018 electronic survey targeted at Finnish forest machine entrepreneurs and forest machine operators who responded anonymously. The questionnaire was sent to 960 entrepreneurs whose contact information was obtained from the member register of the Trade Association of Finnish Forestry and Earth Moving Contractors (TAFFEC) and 890 drivers whose contact information was obtained from the member register of the Finnish Industrial Union.

The study involved 322 people, of whom 87 were entrepreneurs and 235 were drivers (Table 1). The response rate for entrepreneurs was 9.1 and for drivers 26.4. The overall response rate was 17%. Of the entrepreneurs, 80% and of the drivers, 47% were from companies with 1–9 employees. Of the respondents who were from companies with more than ten employees, 20% were entrepreneurs and 53% were drivers, respectively. Overall, 44% of the respondents worked in large logging companies (regional companies). Small businesses represented in the data included subcontractors, self-employed persons, and other forms of corporate organization. According to the geographical distribution, about 14% of all respondents worked in Southern Finland, 30% in Western Finland, 32% in Eastern Finland and about 25% in Northern Finland. The regional distribution of respondents corresponded well to the relative regional distribution of Finnish industrial logging at the time of the survey. The mean age of all participants was 46.1 years (range 21–77 years). The average age of entrepreneurs was 49.4 years (range 28–77 years) and the average age of drivers was 44.8 years (range 21–66 years). Of the entrepreneurs 36% and of the drivers 67% had a formal degree in forestry. About half of the entrepreneurs and about a quarter of the drivers had learned their professional skills on the job. The average number of years in the forestry sector was 26 years for entrepreneurs and 20 years for drivers.

**Table 1.** Background data of the study population *n* = 322.

| Background Variable | % | *n* |
|---|---|---|
| Work schedule | | |
| Regular daily work (06.00–18.00) | 40.4 | 130 |
| Regular two-shift work (06.00–23.00) | 45.0 | 145 |
| Other form of working time | 12.7 | 41 |
| Missing information | 1.9 | 6 |
| Total working time (h/week) | | |
| 40 or under | 20.5 | 66 |
| 41–50 | 59.9 | 193 |
| More than 50 | 19.6 | 63 |
| Distribution of participants by company size | | |
| Companies with 1–9 employees | 50.9 | 164 |
| Companies with 10–20 employees | 49.1 | 158 |
| Distribution of participants by type of company | | |
| Large companies (Regional companies) | 43.5 | 140 |
| Subcontractors | 25.1 | 81 |
| Self-employed persons | 25.8 | 83 |
| Other forms of enterprise | 5.6 | 18 |
| Work experience in the forestry sector (years) | | |
| 1–10 | 20.2 | 65 |
| 11–20 | 30.1 | 97 |
| More than 20 | 49.7 | 160 |

**Table 1.** *Cont.*

| Background Variable | % | *n* |
|---|---|---|
| Education | | |
| Degree in forestry | 59.3 | 191 |
| Learned at work | 30.5 | 98 |
| Other vocational training | 9.0 | 29 |
| Missing information | 1.2 | 4 |

### 2.2. Methods

Perceived work ability was determined by the questions (1) "What score do you currently give to your work ability compared to your lifetime best?" and (2) "to work ability in relation to the mental demands of the job". The questions were rated on a scale from 0 to 10 (0 = very poor, 10 = very good) [48,49]. The questions describing work ability selected for the study have been assessed as valid when looking at subjective work ability in relation to work [50–52]. For the logistic regression analysis, the work ability variables (scale 0–10) were dichotomized (0 = decreased work ability and 1 = good work ability), so that, good work ability scored 8–10 and decreased work ability 0–7. The classification was based on a previous study of a general Finnish population [48].

Factors related to work ability were defined as job resource factors based on the Work Ability House model [34]. In our study, we looked at work resources, such as work organization, opportunities to influence one's own work, employment security, and the meaningfulness of work (experience of the wider social significance of work in the value chain of sustainable development of the forest industry) [40,53] and coworker support, as well as the experience of fairness at work [54–56]. In addition, access to professional development is one example of a task-related resource that was included in our study [29,38].

The work resource factors employed followed those used in previous studies and were classified as dichotomous for statistical analysis (Table 2) [39,53,57]. Individual missing values in other variables were replaced by a group mean or median according to the scale of the variable, to maximize the amount of data.

**Table 2.** Work resource factors.

| Work Resource Factors | Question, Scales and Categories |
|---|---|
| Work organization | How well is your work organized? Scale 0–10. 0–7 poorly, 8–10 well |
| Support from colleagues at work | Do you get support from your colleagues in difficult work situations? Scale 0–10. 0–7 poorly, 8–10 well |
| Experience of fairness at work | Are you treated fairly in your workplace? Scale 0–10. 0–7 poorly, 8–10 well |
| Opportunities to learn new things and skills | Are there opportunities to learn new things and skills in your own work? Scale 0–10. 0–7 poor, 8–10 good |
| Work-related training | Do you get enough training to support your job? Scale 0–10. 0–7 insufficient, 8–10 sufficient |
| Meaningfulness of work | Do you feel that you are doing meaningful work? Scale 1–6. 1–5 = rarely, 6 = daily |
| Opportunities to influence one's own work | Do you have opportunities to influence your work and working conditions? Scale 1–4. 1–3 = rarely, 4 = often |
| Employment security | What is causing you uncertainty about the future? Work continuity/predictability. Scale 1–5. 1–4 = uncertain, 5 = certain |

The associations between job resources and work ability, and the statistical significance of these associations were examined using the Mann-Whitney U-test (significance level:

$p \leq 0.05$). Logistic regression was used to examine the relationship between independent job resource factors and dependent work ability factors. Relationships were interpreted using odds ratios (ORs) and their 95% confidence intervals. SPSS 25.0 was used to analyze the data. The research of which this study is a sub-project has been approved by the Tampere region Ethics Committee for the Humanities (ethics approval statement 15/2016).

## 3. Results

There was no significant difference in the work ability of entrepreneurs and drivers. Work ability can be defined as good on average, as well as when compared to the lifetime best and the mental demands of the job. Among resources, the organization of work, the experience of justice at work and the opportunity to learn new things, as well as the perceived relevance of one's work and the opportunities to influence one's own work, differed statistically significantly between entrepreneurs and drivers. However, the average level of resources was good in both occupational groups (Table 3).

**Table 3.** Mean of work ability factors and work resource factors. Differences between entrepreneurs and drivers, *p*-value.

| Work Ability Factors and Work Resource Factors | Total Sample (*n* = 322) | | Entrepreneurs (*n* = 87) | | Drivers (*n* = 235) | | Entrepreneurs vs. Drivers |
|---|---|---|---|---|---|---|---|
| | Mean | sd | Mean | sd | Mean | sd | *p*-Value |
| **Work ability factors** | | | | | | | |
| Work ability compared to your lifetime best (scale 0–10) | 7.3 | 2.2 | 7.0 | 2.2 | 7.3 | 2.2 | 0.233 |
| Work ability in relation to the mental demands of the job (scale 0–10) | 7.9 | 2.1 | 8.0 | 1.9 | 7.8 | 2.2 | 0.942 |
| **Work resource factors** | | | | | | | |
| Work organization (scale 0–10) | 7.4 | 2.1 | 8.0 | 1.2 | 7.1 | 2.3 | 0.005 |
| Support from colleagues at work (scale 0–10) | 8.3 | 1.9 | 8.4 | 1.4 | 8.3 | 2.0 | 0.206 |
| Experience of fairness at work (scale 0–10) | 7.8 | 2.3 | 8.6 | 1.4 | 7.5 | 2.5 | 0.001 |
| Opportunity to learn new things and skills (scale 0–10) | 7.7 | 2.4 | 8.7 | 1.3 | 7.4 | 2.6 | <0.001 |
| Work-related training (scale 0–10) | 7.5 | 2.4 | 8.1 | 1.4 | 7.3 | 2.7 | 0.142 |
| Meaningfulness of work (scale 1–6) | 4.9 | 1.4 | 5.3 | 1.1 | 4.7 | 1.4 | 0.001 |
| Opportunities to influence one's own work (scale 1–4) | 3.2 | 0.8 | 3.6 | 0.6 | 3.0 | 0.9 | <0.001 |
| Employment security (scale 1–5) | 2.4 | 1.1 | 2.3 | 1.1 | 2.4 | 1.1 | 0.251 |

Good work organization, strong social support and perceived fair treatment at work, as well as the opportunity to maintain professional skills, are resources associated with good work ability. Two-thirds of those working in large logging companies (regional companies) and more than half of those working in other businesses felt that their work was well organized. More than 80% of all respondents reported that they received strong social support from a co-worker. According to all respondents, there is likely to be good work ability in relation to mental work demands when work is well organized (OR 2.76, 95% CI 1.70–4.49). In particular, good social support from co-workers (OR 4.06, 95% CI 2.24–7.36) and perceived fair treatment (OR 4.62, 95% CI 2.77–7.70) were related to good work ability in relation to mental work demands. Across the whole material, meaningful work was related to good mental work ability (OR 2.61, 95% CI 1.56–4.36). The same was true for the ability to influence one's own work and mental work ability (OR 2.32, 95% CI 1.40 to 3.85). More than 40% of respondents stated that their work was meaningful on a daily basis and almost half of the respondents felt that they had the opportunity to influence their work. Among the participants surveyed, 70% had experienced fair treatment in their work. 70% of study participants experienced the opportunity to maintain professional

abilities. Nearly 60% of respondents felt that their work was continuous and safe. Job security was related to good work ability in relation to the requirements of mental work (OR 1.70, 95% CI 1.04–2.77). Throughout the data, all work resources were associated with good work ability, especially when work ability was assessed in relation to mental work demands. The association was very significant.

The likelihood of good work ability comparing well to lifetime best is significantly related to good work organization, co-workers' social support, experience of justice at work, opportunities to learn new things and to be trained at work. The association was very significant. In contrast, in this context, the relevance or impact of work does not increase the likelihood of good work ability (Table 4).

**Table 4.** Association between good work resource factors and good work ability (level 8–10, on scale 0–10) (logistic regression analysis). Total data *n* = 322.

| Good Work Resource Factors | Work Ability Compared to Lifetime Best | | Work Ability in Relation to Mental Work Demands | |
|---|---|---|---|---|
| Total Data *n* = 322 | OR | 95% CI | OR | 95% CI |
| Work organization | 1.97 | 1.25–3.11 | 2.76 | 1.70–4.49 |
| Support from colleagues at work | 2.65 | 1.46–4.81 | 4.06 | 2.24–7.36 |
| Experience of fairness at work | 2.31 | 1.42–3.74 | 4.62 | 2.77–7.70 |
| Opportunity to learn new things and skills | 2.70 | 1.65–4.40 | 2.97 | 1.80–4.91 |
| Work-related training | 2.45 | 1.51–3.98 | 2.72 | 1.65–4.48 |
| Meaningful work on a daily basis | 1.41 | 0.90–2.21 | 2.61 | 1.56–4.36 |
| Opportunities to influence one's own work | 1.30 | 0.83–2.03 | 2.32 | 1.40–3.85 |
| Employment security | 1.67 | 1.07–2.61 | 1.70 | 1.04–2.77 |
| ref. = poor work resources | | | | |

Good work organization, strong social support and perceived fair treatment at work, as well as the ability to maintain skills, are labor resources related to both entrepreneurs and drivers when looking at work ability, especially in terms of mental demands at work (Table 5). 76% of entrepreneurs felt that their work was well organized, as did 55% of drivers. More than 80% of entrepreneurs and drivers received strong social support from colleagues. Forest machine drivers are more likely to have good work ability compared to their lifetime best when work is well organized (OR 2.03, 95% CI 1.20 to 3.43). For entrepreneurs, the difference was not significant (OR 2.21, 95% CI 0.81–6.04). In addition, good social support from co-workers (OR 2.42, 95% CI 1.24 to 4.72) and perceived fair treatment (OR 2.29, 95% CI 1.33 to 3.93) were related to drivers' good ability to work when compared to their lifetime best. Similarly, fair treatment was not significant for entrepreneurs (OR 3.39, CI 0.97–11.81). In contrast, fair treatment at work in terms of mental requirements significantly increased the likelihood of good working capacity for entrepreneurs (OR 5.62, CI 1.68–18.81). Opportunities for professional development and perceived meaningfulness of work are also resource factors related to good working capacity for drivers. More than half (53%) of entrepreneurs and 38% of drivers found their work to be meaningful on a daily basis. The relevance of the work was particularly related to drivers' when assessing work ability in relation to the mental requirements of the work (OR 2.56, CI 1.40–4.70). For entrepreneurs, good work organization, social support from colleagues and experience of justice, as well as opportunities to learn new things and influence one's own work, were significant in terms of mental work demands. For drivers, work-based training is relevant to the mental demands of the job. Correspondingly, this is not relevant to entrepreneurs' work ability (Table 5).

**Table 5.** Association between good work resource factors and good perceived work ability (level 8–10, on scale 0–10) in occupational groups of drivers and entrepreneurs (Logistic regression analysis).

| Good Work Resource Factors | Work Ability Compared to Your Lifetime Best | | Work Ability in Relation to Mental Work Demands | |
|---|---|---|---|---|
| **Drivers *n* = 235** | **OR** | **95% CI** | **OR** | **95% CI** |
| Work organization | 2.03 | 1.20–3.43 | 2.39 | 1.37–4.16 |
| Support from colleagues at work | 2.42 | 1.24–4.72 | 3.63 | 1.86–7.11 |
| Experience of fairness at work | 2.29 | 1.33–3.93 | 4.35 | 2.44–7.76 |
| Opportunity to learn new things and skills | 2.45 | 1.43–4.24 | 2.46 | 1.41–4.31 |
| Work-related training | 2.37 | 1.37–4.11 | 2.65 | 1.51–4.67 |
| Meaningful work on a daily basis | 1.58 | 0.92–2.70 | 2.56 | 1.40–4.70 |
| Opportunities to influence one's own work | 1.24 | 0.72–2.13 | 1.79 | 0.98–3.27 |
| Employment security | 1.44 | 0.85–2.42 | 1.46 | 0.84–2.53 |
| **Entrepreneurs *n* = 87** | | | | |
| Work organization | 2.21 | 0.81–6.04 | 4.09 | 1.41–11.81 |
| Support from colleagues at work | 4.03 | 1.01–16.10 | 5.60 | 1.56–20.13 |
| Experience of fairness at work | 3.39 | 0.97–11.81 | 5.62 | 1.68–18.81 |
| Opportunity to learn new things and skills | 14.52 | 1.71–119.25 | 7.12 | 1.84–27.51 |
| Work-related training | 3.39 | 1.08–10.68 | 2.57 | 0.84–7.88 |
| Meaningful work on a daily basis | 1.13 | 0.49–2.64 | 2.46 | 0.91–6.69 |
| Opportunities to influence one's own work | 1.80 | 0.74–4.39 | 4.08 | 1.48–11.24 |
| Employment security | 2.50 | 1.02–6.16 | 3.62 | 1.10–11.90 |
| ref. = poor work resources | | | | |

## 4. Discussion

The study showed that good work resources increased the likelihood of good work ability. The results confirm the importance of work resources in defining work ability. As a social phenomenon, work ability is broader than merely medical work ability [29,58,59]. Focusing on strengthening work resources may support work ability and enable participation in productive work, despite disability or medical and social constraints [60]. In this study, good work organization was an example of how work resources were associated with good work ability. This supports the findings from previous studies on how good work organization strengthens employees' ability to adapt flexibly to changes in a competitive environment [11,43,61]. According to the present study, social support from co-workers and perceived fair treatment at work are associated with good work ability. Similar results on the importance of social support have been reported by Peters et al. [44] and on the importance of fair treatment on work ability by Spanier et al. [62]. Good professional competence was a resource that was linked to good work ability. Professional competence is a work resource but also a work requirement. At the same time, challenging work that requires skill is also motivating to the employee. From the company viewpoint, professional competence also supports work productivity [63]. The digitally controlled work process and the automation of forest machines have increased the competence requirements of forestry work. Technological development requires a constant learning of new skills [19]. In this study, two thirds of the respondents perceived the upkeep of professional competence to be well-organized in their own workplace.

The study results confirm that the perceived meaningfulness of one's work is a work resource that supports the well-being of employees [40,64], as are opportunities to influence one's own work [65]. Influence on one's own work and involvement in job development are organizational support measures that support the experience of job security [66]. In our study, job security increased the probability of good work ability. Similarly, the link between job security and employee well-being has been previously reported by Horowitz [67].

In this study, social sustainability was addressed from the perspective of work ability and work resources This is in line with Di Fabio's [68] observations, related to the psychology of sustainability development within organizations. The starting point was the

internationalization of the forest industry and the substantive change in timber harvesting work which has changed the responsibilities and importance of work in the forest industry value chain [13]. Therefore, it is also relevant to see timber harvesting as part of an international market economy and, on the other hand, as socially sustainable forestry. Extensive timber harvesting work is at the heart of the sustainable development of the forest industry value chain. The competence requirements of the work include knowledge of environmental legislation, an example of which is the protection of biotopes in connection with felling work. Socio-technical changes have required forest machine entrepreneurs and employees to adapt in terms of content and organization to a new operating environment, marked by uncertainty [4,5]. Digitization has significantly changed the responsibilities of logging work, the requirements for work efficiency are being emphasized and the work has become cognitively stressful [5]. Tengland [29] has drawn attention to the changing characteristics of work that affect work ability. This view is supported by the efficiency requirements of logging work, the subjectivity of work, and the increase in personal responsibility for results which increases the possibility of presenteeism and hides a probable work ability risk. In forestry work, as in many other fields, a competitive and efficient employee, capable of meeting performance targets, is in demand. This study looks at changes in society, the forest industry and timber harvesting work, and, against this background, examines the ability of employees to work, their opportunities to influence their own work and their development at work [65,69]. The study gives indications that socially sustainable development is not being achieved as desired in timber harvesting. Examples of this are competition and work efficiency requirements, as well as time pressures and job insecurity. Responsibilities for work performance are placed on the individual among both entrepreneurs and employees.

Previous studies have found that work resources can be increased and workers' ability to work supported by systematic measures in the workplace [34,70]. Workplace interventions to facilitate work are one solution to safeguard an employee's resources when health or ability to work is impaired [71]. Strengthening resources through work development and modification can support workers' ability to adapt to substantive and technological changes [65,69]. In contrast Oakman et al. [72] have reported that traditional workplace health promotion activities, both individual and wider-ranging varieties, such as exercise programs, have provided only little evidence of effects on work ability and further systematic research is needed on what measures have an impact.

Our results support the findings from previous studies that co-worker social support and opportunities to influence one's own work buffer work demands and support work ability [37,39,57,73]. Boström et al. [32] have also reported that adequate opportunities to influence one's own work support work ability. The links between work resources and good work ability were independent of organizational status, as Demerouti et al. [42] have previously observed. Anticipating changes in work and identifying the effects of changes are essential in promoting work ability [29].

In the global restructuring of the forest industry, the reorganized and wide-ranging remit of the timber harvesting sector has changed the responsibilities and content of the work. Digitization and technological developments in timber harvesting have increased competence requirements and changed the forms of work organization. As a result of these changes, work demands and work ability must be assessed in new ways. This emphasizes the importance of work resources, both for the individual's work ability, as well as for the company's successful operation [63,74]. By strengthening work resources and developing work, the organization's ability to support the work ability of its personnel can be increased as the work changes. By increasing the opportunities for employees to influence and modify their own work, work resources can be strengthened [65,69,75].

*Strengths and Limitations of the Study*

When evaluating our research results, the cross-sectional study design must be taken into account, i.e., cause-and-effect relationships cannot be determined on the basis of this data. The generalizability of the survey results is weakened by the low overall response

rate (17%). Similar response rates from forestry surveys have been reported in Finland and Ireland [76]. On the other hand, the generalizability of the survey results is supported by the geographically even relative distribution of the respondents throughout Finland. The participants in the study represented clearly the focus areas for industrial logging in Finland in relation to the regional volumes of industrial wood harvesting in 2018. The data included representatives of all actors in the Finnish timber harvesting industry and companies of different sizes, as well as key company types, regional entrepreneurs, subcontractors and independent entrepreneurs. Among drivers, the age groups 18–30 years and over 50 years were slightly underrepresented. The age distribution and work experience of the drivers involved corresponded well to the age distribution of the general population.

## 5. Conclusions

Our study showed that good work resources increased the likelihood of good work ability. New work forms, as well as changes in work content and professional demands, may increase the significance of work resources when regarding work ability. Work resources refer to the physical, mental, social or organizational characteristics of work, such as work organization, social support at work and opportunities to influence one's own work. As work life changes, companies would do well to pay special attention to bolstering work resources as a part of the company's daily management.

Work ability as a social construct and work resources must be considered in connection with the operating environment in which the work is done. According to the multidimensional concept of work ability, it is manifested in relation to changes in work. Therefore, the new requirements brought about by the change in working life must be taken into account when striving for effective actions. Based on this study, the solution to supporting and maintaining work ability is the proactive strengthening of work resources in companies and workplaces.

The results of our study call for further research to investigate new forms of work organization and to identify work-related resources that effectively buffer the demands of these new forms of work. Societal changes, and changes in the competitive environment of work and business place new demands on work ability. In the workplace, new systematic solutions are needed to support work ability and promote productive operation.

**Author Contributions:** Conceptualization, H.P., A.S. and C.-H.N.; Formal analysis, H.H.; Supervision, C.-H.N.; Writing—original draft, H.P.; Writing—review & editing, A.S., M.S., H.K., H.H. and C.-H.N. All authors have read and agreed to the published version of the manuscript.

**Funding:** This research was funded by Finnish Labour Protection Fund, grant number 115082 and Metsämiesten Säätiö Foundation.

**Institutional Review Board Statement:** The study was conducted according to the guidelines of the Declaration of Helsinki and approved by the Ethics Committee of University of Tampere (protocol code 15/2016).

**Informed Consent Statement:** Informed consent was obtained from all subjects involved in the study.

**Data Availability Statement:** The data presented in this study are available on request from the corresponding author. The data are not publicly available due to [The study is a part of the dissertation research].

**Conflicts of Interest:** The authors declare no conflict of interest.

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
