# Peer review of "Associations between Work Resources and Work Ability among Forestry Professionals"

_sustainability, doi:10.3390/su13094822_

Round 1
Reviewer 1 Report
Regarding to paper, for me, it doesn't add new information on topic because in literature there are already many studies that show how much effective work organization and social support from co-workers were resources that increased the likelihood of good work ability.
The authors can investigate which organizational variable increases or decreases the work ability in this profession.
Moreover, the authors don't use the work ability index to investigate the levels of work ability in workers.
Author Response
see attachment 1

Reviewer 2 Report
This article treats the working conditions of Finland 's forestry workers in terms of work ability and efficiency. The authors preformed a questionnaire based survey among entrepreneurs and machine operators in this important segment of Finland economy to obtain information about the significance of specific working conditions and to quantify their impact on perceived working ability.
The article is well written, I did only spot two spelling errors (see below). The study has been performed carefully, the database is solid and statistics are well described and sufficient to support the found differences. Citations and literature are fine. To make it short, from my perspective there is little to criticize about the scientific quality of this paper.
However, it has to be mentioned that aside of the good scientific quality, there are two points that might be of relevance for a possible publication.
First, the study investigated the significance of working conditions for working ability where the general outcome is quite expected (see the conclusion line at 208-209). There is of course additional data about relative impact of specific resource factors and the overall situation in Finland that might be of interest for a rather small part of the audience.
Second, but related to the first point, is the scope of this study. In this manuscript no environmental topics or aspects of natural sustainability are touched. A possible social sustainability indicator may be seen described by the current situation of the forestry workers in Finland. However, this is not a predominantly social study, but rather an investigation of economic sustainability of an industry segment. While this is not a problem from a scientific view, when checking the subject areas of Sustainability only little overlap with the aims and scope of the journal is found. I have to leave it to the editor, if progress and stability in an industrial business segment is considered a form of sustainability in the view of the journal.
Apart from these rather editorial problems, I can recommend this paper for publication when only considering quality related arguments.
Minor points:
The authors should explain the term "meaningfulness" as used in the study. This term may be either understood like an instruction that makes sense in terms of fulfilling the specific working task or, in a much wider context, if the activity is meaningful in a social or natural context.
Line 236-237: "The term "job accommodation" needs to be explained and/or changed. The term is not mentioned in citation 69.
Spelling:
Table 1: "schedule"
Table 2: "colleagues"
In my copy of the PDF-file there are several spacing errors (Lines: 157, 159, 177
239: "In contrast..." better than "Instead"
Author Response
see attachment 2

Reviewer 3 Report
Article
ASSOCIATIONS BETWEEN WORK RESOURCES AND WORK ABILITY AMONG FORESTRY PROFESSIONALS
Dear authors,
The paper is a very interesting presentation but still lucking a bit more health status approach to work ability. It is correct that work characteristics are important to define workability – but beside other factors where health status is very much important indeed. Following the variables described in the article the conclusions are correct but nevertheless at least in the Discussion it should be mentioned that sickness presenteeism and its danger for heath can be hidden behind the frames of good work ability in a social concept of this term!
There are differences between entrepreneurs and drivers, and also Table 4 should be presented the same way as Table 5 to give better presentation. There must be more significant differences between the two groups observed as presented in a text.
By the way – there is probably a mistake overlooked:
- Table 1: there is a word YLI stated before 20 (years of work experience in forestry sector) – not translated to English?
Best wishes!
Thank you to let me know the added Discussion part.
Author Response
see attachment 3

Round 2
Reviewer 1 Report
My vote is average. The paper is improved.